# Development and Validation of an Insulin Resistance Model for a Population with Chronic Kidney Disease Using a Machine Learning Approach

**DOI:** 10.3390/nu14142832

**Published:** 2022-07-09

**Authors:** Chia-Lin Lee, Wei-Ju Liu, Shang-Feng Tsai

**Affiliations:** 1Division of Endocrinology and Metabolism, Department of Internal Medicine, Taichung Veterans General Hospital, Taichung 407219, Taiwan; u502107@yahoo.com.tw; 2Department of Medical Research, Taichung Veterans General Hospital, Taichung 407219, Taiwan; u103092002@cmu.edu.tw; 3Department of Public Health, College of Public Health, China Medical University, Taichung 406040, Taiwan; 4School of Medicine, National Yang-Ming University, Taipei 112304, Taiwan; 5Department of Post-Baccalaureate Medicine, College of Medicine, National Chung Hsing University, Taichung 402204, Taiwan; 6Division of Nephrology, Department of Internal Medicine, Taichung Veterans General Hospital, Taichung 407219, Taiwan; 7Department of Life Science, Tunghai University, Taichung 407224, Taiwan

**Keywords:** insulin resistance, HOMA-IR, chronic kidney disease, artificial intelligence, machine learning, deep learning, national health and nutrition examination survey (NHANES)

## Abstract

Background: Chronic kidney disease (CKD) is a complex syndrome without a definitive treatment. For these patients, insulin resistance (IR) is associated with worse renal and patient outcomes. Until now, no predictive model using machine learning (ML) has been reported on IR in CKD patients. Methods: The CKD population studied was based on results from the National Health and Nutrition Examination Survey (NHANES) of the USA from 1999 to 2012. The homeostasis model assessment of IR (HOMA-IR) was used to assess insulin resistance. We began the model building process via the ML algorithm (random forest (RF), eXtreme Gradient Boosting (XGboost), logistic regression algorithms, and deep neural learning (DNN)). We compared different receiver operating characteristic (ROC) curves from different algorithms. Finally, we used SHAP values (SHapley Additive exPlanations) to explain how the different ML models worked. Results: In this study population, 71,916 participants were enrolled. Finally, we analyzed 1,229 of these participants. Their data were segregated into the IR group (HOMA IR > 3, n = 572) or non-IR group (HOMR IR ≤ 3, n = 657). In the validation group, RF had a higher accuracy (0.77), specificity (0.81), PPV (0.77), and NPV (0.77). In the test group, XGboost had a higher AUC of ROC (0.78). In addition, XGBoost also had a higher accuracy (0.7) and NPV (0.71). RF had a higher accuracy (0.7), specificity (0.78), and PPV (0.7). In the RF algorithm, the body mass index had a much larger impact on IR (0.1654), followed by triglyceride (0.0117), the daily calorie intake (0.0602), blood HDL value (0.0587), and age (0.0446). As for the SHAP value, in the RF algorithm, almost all features were well separated to show a positive or negative association with IR. Conclusion: This was the first study using ML to predict IR in patients with CKD. Our results showed that the RF algorithm had the best AUC of ROC and the best SHAP value differentiation. This was also the first study that included both macronutrients and micronutrients. We concluded that ML algorithms, particularly RF, can help determine risk factors and predict IR in patients with CKD.

## 1. Introduction

Chronic kidney disease (CKD) is defined by the presence of renal damage or reduced renal function lasting for at least three months, irrespective of the cause. The care of CKD relies on avoiding further renal injury, and the treatment of complications such as anemia, metabolic acidosis, secondary hyperparathyroidism, edema, hyperkalemia, and nausea or vomiting. In 2017, a global burden of disease study reported that CKD had caused 1.2 million deaths worldwide and was the world’s 12th leading cause of death [1]. Global all-age CKD mortality also increased 41.5% from 1990 to 2017 [1]. A recent analysis suggested that global CKD prevalence in 2017 was 9.1% (697.5 million cases) [2]. In Taiwan, the national prevalence of CKD is even as high as 11.93% according to a prospective cohort study based on 462,293 adults [3]. The global expansion of this disease is mainly driven by the increasing prevalence of a number of disorders, such as diabetes mellitus, hypertension, obesity, and aging [4]. The progression of CKD has serious consequences [5], such as high risk of mortality, end-stage kidney disease (ESKD), mineral bone disease, accelerated cardiovascular disease (CVD), and infections. In a cohort study of 462,293 individuals [5], patients with CKD had a mortality 83% higher than the all-cause mortality (hazard ratio (HR) 1.83, 95% CI 1.73–1.93) and 100% higher than CVD (HR 2.00, 95% CI 1.78–2.25). Despite the attention of clinicians, CKD care remains suboptimal, ranking the prevalence and incidence of ESRD as top one or top two worldwide.

Insulin resistance (IR) is the attenuated effect of insulin in cells of liver, skeletal muscle, or adipose tissue. IR is considered a status in which a given concentration of insulin is associated with a subnormal glucose response [6]. In clinical practice, static testing is the homeostatic model for assessing IR (or HOMA-IR). HOMA-IR estimates insulin sensitivity with an equation, including the fasting insulin–glucose product divided by a constant [7]. Once an individual develops IR, compensatory insulin secretions increase. IR has a number of underlying causes [8], including obesity, stress, medication (e.g., steroids), pregnancy, insulin antibodies, and genetic defects in insulin signaling pathways. IR is well-known as an indicator for the early diagnosis of diabetes mellitus (DM). In addition to DM, IR has also been noticed as a very early alteration in CKD patients [9].

As eGFR falls, IR gradually progresses. Until recently, only 11 studies had been reported on the epidemiology of IR in CKD patients [10]. IR is associated inversely with the level of eGFR. The muscle is the primary site for glucose disposal, and IR in uremia depends on peripheral resistance for the action of insulin at the postreceptor level [11]. The etiology of tissue insensitivity to insulin in CKD patients is multifactorial, including genetic problems, background physical activity, as well as CKD risk factors, including inflammation and oxidative stress, adipokine derangements, vitamin D deficiency, metabolic acidosis, anemia, and microbial toxins [10]. Therefore, as renal function deteriorates, IR also becomes worse.

The strong association between IR and atherosclerosis is known for the general population without CKD [12,13]. In a stratified analysis of CKD patients who were smokers or physically inactive, IR predicted a high death risk [14]. In a small cohort of 170 nondiabetic dialysis patients in Japan, IR predicted mortality independently of other risk factors, including inflammation and BMI [15]. IR in CKD was reported to be associated with CVD with a possible causal relationship, as the results of oxidative stress [10], chronic inflammation [10], and endothelial injury [16]. The association was considered as strong according to reported studies [15,17]. In addition, IR may also cause the progression of CKD [18,19].

Therefore, the early prediction of IR in patients with CKD should be mandatory to avoid further CVD and functional deteriorations. However, there are few studies regarding the prediction of IR in CKD patients. The main difficulties of these studies are the complicated underlying background status, physical activity, diet intake, and disease status of CKD. Sophisticated computation and machine learning (ML) provide a new opportunity for dealing with such a complicated status for predictive models. Growing attention has been given recently to the use of the ML approach in this regard. Thus, here, we used differential models of ML to develop predictive models regarding IR in patients with CKD.

## 2. Materials and Methods

### 2.1. Data Source: National Health and Nutrition Examination Survey (NHANES)

The National Health and Nutrition Examination Survey (NHANES) is one of a series of health-related programs in the USA conducted periodically by the Centers for Disease Control (CDC) and Prevention’s National Center for Health Statistics (NCHS). Their data are released to the public for free. Our study protocol was approved by the research ethics review board at the NCHS, and all participants or proxies provided written informed consent. This large ongoing dietary survey was conducted to cross-sectionally assess the health and nutritional status of community-dwelling individuals in the USA. The examinations included anthropometric measurements, questionnaires on health and nutrition, and laboratory testing. Participants completed in-home interviews. We analyzed participants in the NHANES from 1999 to 2012. Participants were excluded from analyses if they aged <18 years, had no data on estimated glomerular filtration rate (GFR), or were without complete data on anthropometric measurements, questionnaires, and laboratory tests.

Histories were collected for diabetes mellitus (DM), cardiovascular disease (CVD), smoking [20], and hypertension through specific questionnaires: MCQ160C for CVD, MCQ220 for cancer or malignancies, and DIQ010 for DM [21]. Mortality records were provided by NCHS and NHAES. These surveys were created on a record matching NHANES and the National Death Index (NDI) death certificates, which is an NCHS centralized database of all deaths in the USA from 1979 onward. Causes of death were obtained from the NHANES.

### 2.2. Definition of Target Population and Data Collection

The value of eGFR (mL/min/1.732 m^2^) was calculated according to the chronic kidney disease epidemiology collaboration (CKD-EPI) equation [22]. CKD was defined at eGFR < 60 mL/min/1.732 m^2^. Baseline variables included the following: age, gender, race (non-Hispanic white, non-Hispanic Black, and Mexican American), body mass index (BMI) (weight in kg divided by height in meters), eGFR (mL/min/1.732 m^2^), total cholesterol (TC), high-density lipoprotein (HDL)(mg/dL), triglyceride (mg/dL), and fasting plasma glucose (mg/dL). As for daily food intake, we collected the following data: daily protein intake (gm/kg), daily energy intake (kcal/kg), daily protein intake ratio (%), daily carbohydrate intake ratio (%), daily total fat intake ratio (%), daily fiber intake (gm), daily cholesterol intake, (mg), daily folate intake (mcg), daily total saturated fatty acid (gm), daily total monounsaturated fatty acid (gm), total polyunsaturated fatty acid (gm), vitamin B_6_ (mg), vitamin B_12_ (mcg), vitamin C (mg), calcium (mg), phosphorus (mg), magnesium (mg), iron (mg), zinc (mg), copper (mg), sodium (mg), potassium (mg), selenium (mcg), caffeine (mg), theobromine (mg), and alcohol (gm).

### 2.3. Definition of Insulin Resistance (IR)

We used the homeostasis model assessment of IR (HOMA-IR) for assessing insulin resistance (IR). HOMA-IR is the most common method used to calculate IR. The outcome measure was IR, as indexed using HOMA-IR. Age, sex, race, year of assessment, BMI, and smoking status were covariates. The key potential confounding variable was waist circumference. Waist circumference was also used to test the presence of effect modification [23]. Increased HOMA-IR is known to strongly associate with the development of type 2 DM, statistically independent of impaired glucose tolerance status, obesity, and body fat distribution [24]. Higher values of HOMA-IR were independently associated with the risk of developing prediabetes [25]. HOMA-IR used the following formula to index insulin resistance: fasting plasma insulin (μU/mL) × fasting plasma glucose (mg/dL)/405 [7]. NHANES provided data on a participants’ measures of fasting insulin and fasting glucose, as well as detailed assessment procedures [20,26,27,28,29,30]. HOMA-IR varied with age: peaked at age 13 years in girls and at 15 years in boys. HOMA-IR value of 2.5 is an indicator of IR in adults [31]. However, no consensus was reached regarding cutoff values of HOMA-IR in patients of different disease disorders. For example, in CKD patients, the cutoff value of HOMA-IR varies from study to study, e.g., being 1.23 in a comparative study [32], 2.0 in another study for the evaluation of renal function deterioration [19], and 5.64 in nondiabetic nonobese patients with CKD [33]. HOMA-IR index values have been reported to be higher in predialysis and dialysis patient groups compared with controls [34]. Most patients with high HOMA-IR have higher incidences of IR. In one study regarding reference ranges of HOMA-IR in normal-weight and obese young Caucasians [35], if HOMA-IR cutoff point was 3.02, the AUROC was 0.73 (95% CI = 0.70–0.75) with 46.3% of sensitivity and 86.2% of specificity. Therefore, we set the cutoff value of IR in our CKD patients at values of HOMA-IR >3.

### 2.4. Model Building Process

We randomly selected 60% of the patient population as the training group. The remaining 40% were the testing group. The estimated proportion of IR was ~50% in patients with poor physical fitness.

Target population was obtained in the training group. We used the technique of upsampling or downsampling for sample balance between the target and nontarget populations. Deep neural network (DNN), one of the deep learning methods, was used to estimate the first chosen method. Other traditional methods of ML algorithm, such as random forest (RF), eXtreme Gradient Boosting (XGboost), and logistic regression algorithms, were also used to compare accuracy with the DNN method. After completing the model training, the testing group was applied for validation. In the training group, ROC (receiver operating characteristic) curves from different algorithms were compared. Both the ROC curve and AUC (area under curve) were used for evaluating classification performance of different classifiers [36]. The targeted value of AUC (0.80) suggested that the model was adequate for predicting IR [37,38]. Finally, we used SHAP values (SHapley Additive exPlanations) to explain how different ML models worked [39].

### 2.5. Statistical Analysis

NHANES is a multiple complex survey design. To represent sample-weighted data and the difference between insulin resistance status, the weighted mean +95% confidence interval and weighted percentage were compared by weighted Chi-square test and weighted regression test. Weighted data were calculated according to analytical guidelines [40]. Moreover, original unweighted variables were used to perform model building of machine learning and deep learning. For unweighted data, continuous variables were reported as means ± SD and categorical data as numbers (percentages). Differences in clinical variables between insulin resistance statuses were tested by using the Chi-square test for categorical variables or paired t-test for continuous variables. All reported p-values were two-sided and considered significant with *p* < 0.05. All statistical analyses were performed using SAS for Windows (version 9.4; SAS, Cary, NC, USA). Deep learning algorithms and other ML (including XGBoost, random forest, and DNN) were conducted with Keras (version 2.4.0), TensorFlow (version 1.10.0), and Python (version 3.6.5).

This study was approved by the Ethics Committee of Taichung Veterans General Hospital, IRB number: CE20023A-2. Moreover, all methods were performed in accordance with the relevant guidelines and regulations.

## 3. Results

In this study, 71,916 participants were enrolled at the first stage. After exclusion (32,295 participants due to ages <18 years, 23,008 participants due to incomplete laboratory data, 634 participants due to incomplete nutrition data, and 14,750 due to being non-CKD participants), we had 1229 participants for the final analysis (Figure 1). Of all participants with CKD, we randomly separated 675 participants for the train group and 185 for the validation group. In the training group, we trained the model via XGBoost, random forest, logistic regression, and DNN. AUCs were finally compared. A total of 369 participants was included in the testing group for model evaluation.

### 3.1. Baseline Characteristics of Participants with Chronic Kidney Disease According to HOMA-IR

Detailed data according to training, validation, and testing groups are shown in Appendix A. In Table 1, data (*n* = 1229) were separated into the IR group (HOMA IR > 3, *n* = 572) or non-IR group (HOMR IR ≤ 3, *n* = 657). The unweighted data of descriptive statistics are shown in Appendix A. The IR group had higher HOMA-IR values (8.44 (7.5–9.38), *p* < 0.0001), younger age (70.76 (69.79–71.74), *p* < 0.001), fewer non-Hispanic white subjects (79.63%, *p* = 0.0112), and a higher BMI (31.94 (31.24–32.63), *p* < 0.001). As for the laboratory data, the IR group had various lipid profiles (all *p* < 0.0001), including a lower HDL (48.07 (46.57–49.57)), lower total cholesterol (189.11 (184.4–193.83)), and higher triglyceride (180.55 (169.83–191.26)). They also had a higher baseline glycohemoglobin (6.3 (6.18–6.42)) and fasting plasma glucose (125.8 (121.11–130.49)) (both *p* < 0.0001). In terms of macronutrients, the IR group had less daily variations (all *p* < 0.0001), including a lower energy intake (19.54 (18.7–20.37) kca/kg), lower daily carbohydrate intake ratio (48.77 (47.55–49.98)%), lower daily protein intake (0.8(0.75–0.84) gm/kg), higher daily protein intake ratio (16.52 (15.89–17.16)%), higher daily total fat intake ratio (34.71 (33.75–35.67)%), higher total saturated fatty acid intake (22.04 (20.64–23.44) gm), higher total monounsaturated fatty acid (24.38 (22.8–25.95) gm), higher total polyunsaturated fatty acid (14.31 (13.29–15.33) gm), and higher daily cholesterol intake (267.78 (244.03–291.53)). For micronutrients, the IR group had less daily variations (all *p* < 0.0001), including a lower daily fiber intake (13.52 (12.67–14.37) gm), lower total folate intake (332.97 (314.62–351.31) mcg), lower vitamin B_6_ (1.67 (1.55–1.79) mcg), higher vitamin B_12_ (4.74 (4.24–5.23) mcg), more vitamin C (78.31 (69.35–87.26) mg), more calcium (780.24 (731.9–828.58) mg), more phosphorus (1118.1 (1059.64–1176.57) mg), more magnesium (237.49 (225.38–249.61) mg), less iron (13.42 (12.62–14.21) mg), more zinc (10.25 (9.58–10.92) mg), more copper (1.1 (1.03–1.17) mg), more sodium (2922.9 (2765.66–3080.13) mg), more potassium (2398.72 (2271.46–2525.97) mg), more selenium (90.55 (85.22–95.88) mcg), more caffeine (149.9 (131.22–168.58) mg), less theobromine (30.09 (24.48–35.71) mg), and less alcohol (2.72 (1.64–3.8) gm). More of the IR group participants also had past histories of hypertension (80.72%) and DM (48.63%) (both *p* < 0.0001).

### 3.2. Model Comparisons in IR prediction for CKD patients

Table 2 summarizes results of validation and test groups among XGboost, RFs, logistic regression, and DNN. In the validation group (Figure 2B), AUC of ROC were all >0.8 (highest being RF, 0.83). RF had a higher accuracy (0.77), specificity (0.81), PPV (0.77), and NPV (0.77). The sensitivity was highest in DNN (0.74). In the testing group (Figure 2C), all AUCs of ROC were >0.76 (highest being XGboost, 0.78). In addition, XGBoost also had a higher accuracy (0.7) and NPV (0.71). RF had a higher accuracy (0.7), specificity (0.78), and PPV (0.7). DNN had a higher sensitivity (0.66) and NPV (0.71).

### 3.3. Relative Importance of Parameters in XGBoost and Random Forest (RF) Algorithms

The highest AUC score of the ROC curve out of the four algorithms was achieved using XGBoost, with RF being the second highest. For the XGBoost and RF prediction models, their relatively important features were obtained (Figure 3A for XGBoost and Figure 3B for RF). The 32 features these models included were as follows: epidemiological domain (age, gender, and BMI), laboratory data domain (total cholesterol, triglyceride, and HDL), macronutrients domain (daily calorie intake, carbohydrate intake ration, protein intake ratio, protein intake amount, and fat intake ratio), and micronutrients domain (daily cholesterol intake, monounsaturated fatty acid intake, saturated fatty acid intake, selenium, sodium, phosphorus, potassium, zinc, polyunsaturated fatty acid, copper, caffeine, fiber, calcium, vitamin B_12_, vitamin B_6_, iron, vitamin C, magnesium, folate, theobromine, and alcohol). In the XGBoost algorithm (Figure 3A), the BMI had a large impact on IR (0.1262), followed by triglyceride (0.0754), the protein intake ratio (0.0537), age (0.0487), blood total cholesterol value (0.0433), daily cholesterol intake (0.0430), daily calorie intake (0.0429), blood HDL value (0.0424), daily zinc intake (0.0420), and daily saturated fatty acid intake (0.0414). In the RF algorithm (Figure 3B), similarly, the BMI had a large impact on IR (0.1654), followed by triglyceride (0.0117), the daily calorie intake (0.0602), blood HDL value (0.0587), and age (0.0446). These differences contributed to the AUC differences for the XGBoost and RF algorithms.

The relative importance from XGBoost and the RF algorithms did not show a positive and negative association of the selected features with IR (Figure 4A for XGBoost and Figure 4B for RF). Some features were well separated to show a positive or negative association with IR in terms of the SHAP value for the XGBoost algorithm (Figure 4A), including the blood triglyceride value (strongly positive impact), BMI (strongly positive impact), daily calorie intake (medium negative impact), and blood total cholesterol value (medium negative impact). For the RF algorithm (Figure 4B), almost all features were well separated to show a positive or negative association with IR in terms of the SHAP value. In contrast, age for IR was not well discriminated against IR in the RF algorithm. SHAP values for all features are showed in Appendix A for the XGBoost algorithm and in Appendix A for the RF algorithm.

## 4. Discussion

IS and hyperinsulinemia are associated with CKD [41,42,43] and cardiorenal metabolic syndrome [44,45,46]. Various observational studies also reported the association between IR and the development of CKD independent of type 2 DM [15,47,48]. Data from NHANES also suggested a strong relationship between CKD and IR independent of type 2 DM [41,49,50]. Therefore, how to predict IR in patients with CKD is important to clinicians. In a study on a chronic renal insufficiency cohort (CRIC) without DM [51], a multivariable-adjusted analysis showed many independent factors associated with a higher HOMA-IR, including age, no-smoking, BMI, waist circumference, hemoglobin, LDL, HDL, triglyceride, and C-reactive protein. Associated factors related to IR in CKD patients were reported. In addition to the above factors, the micronutrients were well reviewed and their altered levels were associated with the trajectory toward IR, DM, oxidative stress, and provided disease-relevant information [52]. Until now, no study had been published on the association between micronutrients and IR in CKD patients. Our present study was the first one on such an association in a CKD population. Moreover, this was also the first study using AI to develop a predictive model for IR in CKD patients. In a review article [53], principles of ML are considered as building algorithms to support predictive models for the outcome. AI can also introduce a paradigm shift in disease care from conventional treatment strategies to building targeted data-driven individualized care.

For our predictive models in the validation group, RF had the highest AUC of ROC (0.83), specificity (0.81), PPV (0.77), and NPV (0.77). Similarly, in the testing group, the AUC of ROC of the RF algorithm almost had the highest values (0.77), second to the XGboost algorithm (AUC of ROC, 0.78). In general, both MLs (XGBoost and RF) appeared to be better predictive models compared with the logistic regression. However, between XGBoost and RF, we preferred the RF algorithm, because of its easier explanations of feature impacts on the IR, based on the SHAP value. SHAP is an extended Shapley value in cooperative game theory. It is used to calculate contributions of features in ML. The SHAP value has been widely used for evaluating the impacts of contributions of each feature from predictive models, such as the network–pharmacokinetic model [54], extubation failure in intensive care units [55], multivariate molecular diagnostic test [56], and the factors associated with the rapid treatment of sepsis [57]. In our present study, the positive and negative SHAP values could be separated more clearly in the RF algorithm (Figure 4B) than in the XGBoost algorithm (Figure 4A). Detailed information on the SHAP values of all features in the RF algorithm can be found in Appendix A. With increasing values of a feature (increasing pattern), the SHAP value also increased. Such features included the BMI, blood triglyceride, ratio of protein intake, total cholesterol intake, phosphorus intake, magnesium intake, zinc intake, copper intake, and selenium intake. However, with decreasing values of a feature (decreasing pattern), the SHAP value decreased. Such features included the blood HDL, blood total cholesterol, total protein amount, calorie intake, ratio of carbohydrate intake ratio, folate intake, vitamin B6 intake, calcium intake, and caffeine intake. Some features had a “J-shape pattern”, including the saturated fatty acid intake, polyunsaturated fatty acid intake, fat intake ratio, saturated fatty acid intake, monounsaturated fatty acid intake, and polyunsaturated fatty acid intake. Interestingly, all features with a U-shape pattern belonged to fat-related factors. Even if those micronutrients had a low impact of feature importance (Figure 3B), they still had a specific pattern of impact on the IR. This condition cannot be identified using traditional statistical analyses.

In the RF algorithm (Figure 3B), the BMI and blood triglyceride value were important features with a large impact on IR, and they presented an “increasing pattern” (Appendix A). The BMI and blood triglyceride levels were also reported to be associated with IR in the CKD cohort [51]. IR could cause the overproduction of every LDL, and that contributes to hypertriglyceridemia [58], which further induces the progression of renal dysfunction [59]. Similarly, in a prospectively study, serum triglyceride levels were found to be higher in patients who progressed in nephropathy compared with those who did not (median 1.21 (range 0.41–2.96) vs. 0.91 (0.31–11.07) mmol/L; *p* = 0.0037) [60]. The above results were consistent with our present evidence on the importance of the serum triglyceride level on IR development in CKD patients.

Fat-related nutrients (saturated fatty acid intake, polyunsaturated fatty acid intake, fat intake ratio, saturated fatty acid intake, monounsaturated fatty acid intake, and polyunsaturated fatty acid intake) had a J-shape pattern, which indicated too many or too few of the fat-related nutrients were associated with an increased pattern in this CKD cohort. Fatty acid-medicated IR can be found in various organs [61]. In another review article [62], IR in CKD patients was strongly associated with free fatty acid levels. Tumor necrosis factor alpha can activate adipose tissue lipolysis, generating free fatty acid. In muscle cells, free fatty acid may further activate many transcription factors and downstream signal transduction pathways. Finally, this pathway causes IR in CKD patients. In addition, the changes in the plasma free fatty acid level correlated linearly with intramyocellular triglyceride (r = 0.74, *p* < 0.003) [63]. The free fatty acid impairment of insulin sensitivity has been repeatedly reported [64]. The composition of a fatty acid diet could have a significant role in modulating IR. Interestingly, in our study with ML, we found that the intake of very low fatty acid-related nutrients also mildly increased IR. The association between fatty acid and IR in CKD patients was kind of similar to a J shape, rather than a linear increasing pattern. This could be partially explained by malnutrition and inflammation, two important players in the reverse epidemiology of this population [65], and they both counteract insulin sensitivity [66].

In this model with ML, we also found that in our patients, micronutrients were associated with the IR, including an increasing pattern (copper intake and selenium intake), and a decreasing pattern (caffeine intake).

The greater the intake of copper (>1 mg/day), the more significantly IR increased (SHAP value of RF in Appendix A). A study on obese Malaysian adults [67] reported a significant positive association between dietary copper and HOMA-IR with intakes of Cu ≥ 13.4 µg/kg/day, 0.276 (CI = 0.025–0.526; *p* = 0.033). Excess copper intake might create oxidative stress, which further favors the progression of T2DM [68].

The greater the intake of selenium (>50 mg/day), the more significantly IR increased (SHAP value of RF in Appendix A). A cross-sectional study on 5423 middle-aged and elderly Chinese participants [69] reported a strong positive correlation between selenium intake and DM. Other cross-sectional surveys on 8876 adults in the US NHNES also showed a positive correlation of a higher serum selenium value and DM [70,71,72]. A hospital-based case–control study on 847 adults [73] showed that the odd ratios of having DM in the second, third, and fourth selenium quartile groups were 1.24 (95% CI 0.78 to 1.98, *p* > 0.05), 1.90 (95% CI 1.22 to 2.97, *p* < 0.05), and 5.11 (95% CI 3.27 to 8.00, *p* < 0.001), respectively, after being adjusted for age, gender, current smoking, current drinking, and physical activity. The recommended daily requirement of selenium is 55 mg for adults [52], consistent with our findings of no more than 50 mg.

Regarding the caffeine intake, the greater the intake (>200 mg/day), the less there was IR as based on the SHAP value. In a rat model, a chronic caffeine intake reversed aging-induced IR by lowering NEFA production and increasing Glut4 expression in skeletal muscles [74]. Caffeine reduces the production of superoxide and the expression of the receptor of advanced glycation end-product at the nucleus tractus solitarii, though it enhances insulin receptor substrate 1-phosphatidylinositol 3-kinase-Akt-neuronal nitric oxide synthase signaling [75]. In another rat model of a high-fat and high-sucrose diet [76], long-term caffeine intake prevented IR, a result related to a drop in circulating catecholamines. In a US NHNES survey (2009–2010 and 2011–2012), caffeine and its metabolites were found to be positively related to IR and beta cell function [77]. In our predictive model with ML, we also found the importance of caffeine intake on IR.

The strength of this study was its novelty, namely, being the first study to investigate predictive models of IR in patients with CKD via ML. Moreover, compared with other studies on predictive models of AI, this was the first study to have enrolled so many macronutrients and micronutrients. Using SHAP values, the impact of feature importance on IR could be better explained. Importantly, the clinical implication of this study was that our predictive model via ML could provide, on a daily basis, an individual warning in our clinical practice. The IR in CKD was strongly associated with atherosclerosis. After this study, we can benefit from the early prediction of IR in patients with CKD to avoid further CVD and functional deteriorations. In the future, we aim to validate this algorithm in another large database via federated learning. We also plan to correlate the association of IR in CKD to clinical outcomes, including CVD, renal function deterioration, and all-cause mortality via machine learning.

There were several limitations in our study. First, we did not have features of genetic data. Second, data in this study were from cross-sectional surveys only. Third, we did not have enough data to separate patients into different stages of CKD. However, in the current form of the dataset, we believe our predictive model of IR in patients with CKD can explain IR based on features, which can be obtained in clinical practice. In the future, we may need to enroll even more features, such as genetic factors, to train our predictive model better.

## 5. Conclusions

This was the first study using ML to predict IR in patients with CKD. In our study, the RF algorithm had the best AUC of ROC and the best differentiation of SHAP values. This was also the first study including both macronutrients and micronutrients. We concluded that ML algorithms, particularly RF, can help determine risk factors and predict IR in patients with CKD.

## Figures and Tables

**Figure 1 nutrients-14-02832-f001:**
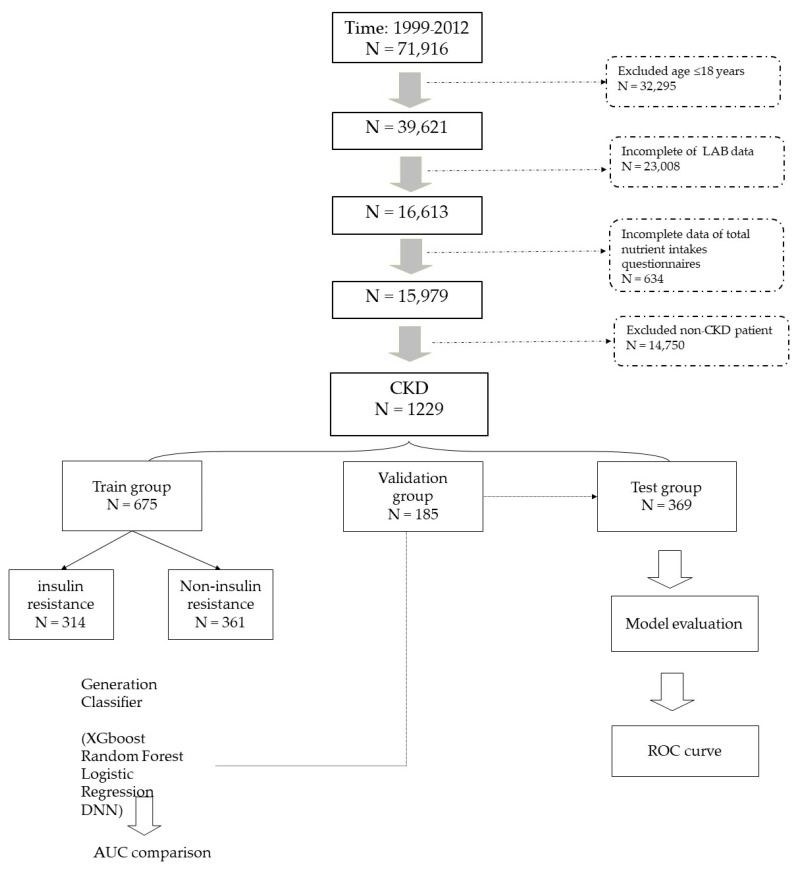
Algorithm of groping in patients with CKD. Initially, all 71,916 participants were enrolled in this study. After exclusion, all 1229 participants with CKD with complete data were analyzed. We separated all 1229 participants into three groups: 675 for train group, 185 for validation group, and 369 for test group. All algorithms were compared via AUC.

**Figure 2 nutrients-14-02832-f002:**
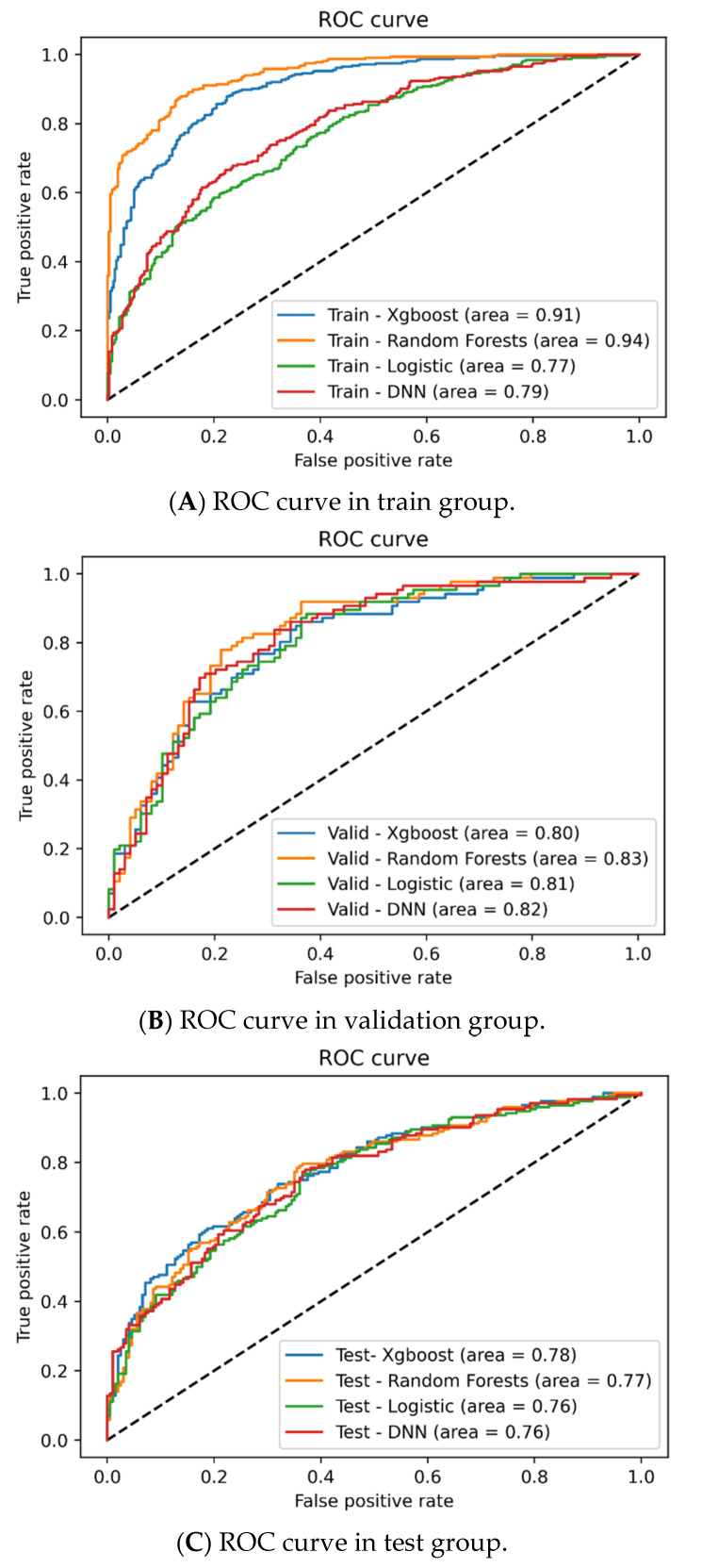
Prediction of insulin resistance in patients with CKD using XGboost, random forest, logistic regression, and DNN in train (**A**), validation (**B**), and test group (**C**). In the train group (**A**), RF had the highest AUC (0.94), followed by XGboost (0.91). In the validation group (**B**), AUCs of ROC were all >0.8 (highest being RF, 0.83). RF had higher accuracy (0.77), specificity (0.81), PPV (0.77), and NPV (0.77). The sensitivity was highest in DNN (0.74). In the testing group (**C**), all AUCs of ROC were >0.76 (highest being XGboost, 0.78). In addition, XGBoost also had higher accuracy (0.7) and NPV (0.71). RF had higher accuracy (0.7), specificity (0.78), and PPV (0.7). DNN had higher sensitivity (0.66) and NPV (0.71).

**Figure 3 nutrients-14-02832-f003:**
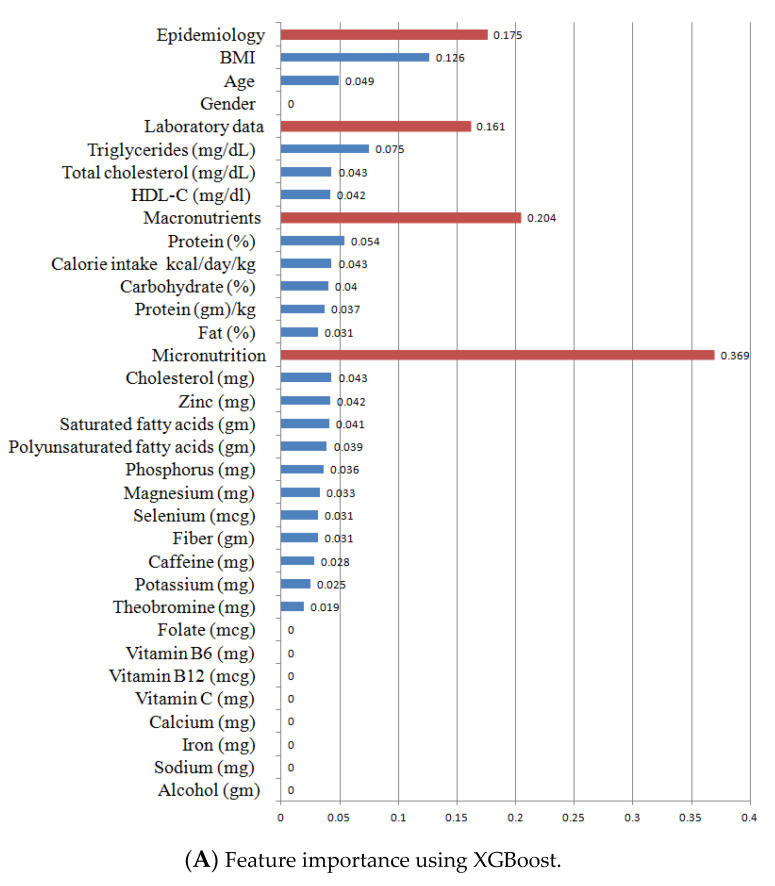
Feature importance. (**A**) XGboost algorithm. (**B**) RF algorithm. In the XGBoost algorithm (**A**), BMI had a large impact on IR (0.1262), followed by triglyceride (0.0754), protein intake ratio (0.0537), age (0.0487), blood total cholesterol value (0.0433), daily cholesterol intake (0.0430), daily calorie intake (0.0429), blood HDL value (0.0424), daily zinc intake (0.0420), and daily saturated fatty acid intake (0.0414). In the RF algorithm (**B**), similarly, BMI had a large impact on IR (0.1654), followed by triglyceride (0.0117), daily calorie intake (0.0602), blood HDL value (0.0587), and age (0.0446). (The dark red line indicated group name and the blue line indicated individual item name).

**Figure 4 nutrients-14-02832-f004:**
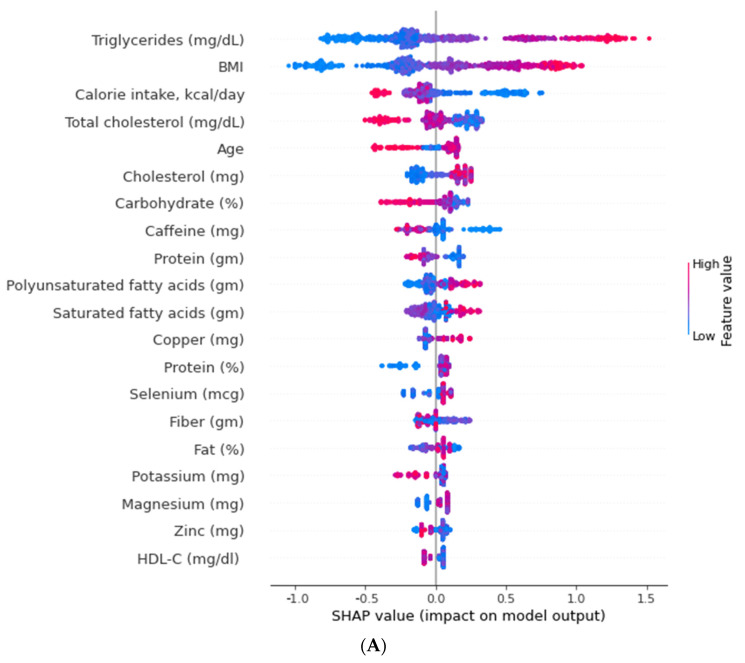
Positive and negative impact explanation of features for predicting insulin resistance using SHAP values. (**A**) for XGBoost and (**B**) for RF. Well separated to show a positive or negative association with IR in terms of SHAP value for the XGBoost algorithm (**A**), including blood triglyceride value (strongly positive impact), BMI (strongly positive impact), daily calorie intake (medium negative impact), and blood total cholesterol value (medium negative impact). For the RF algorithm (**B**), almost all features were well separated to show a positive or negative association with IR in terms of SHAP value. (**A**) Explanation of each feature impact on the IR in the prediction model using the SHAP values in the XGBoost algorithm. (**B**) Explanation of each feature impact on the IR in the prediction model using the SHAP values in the RF algorithm.

**Table 1 nutrients-14-02832-t001:** Baseline characteristics of all participants with chronic kidney disease according to HOMA-IR.

	Overall	HOMA IR ≤ 3	HOMA IR > 3	*p*-Value
Epidemiology				
Case number, n	1229	657	572	
HOMA-IR	4.73(4.21–5.25)	1.76(1.68–1.84)	8.44(7.5–9.38)	<0.0001
Age (year/o)	71.51(70.7–72.33)	72.12(70.91–73.32)	70.76(69.79–71.74)	<0.0001
Male, *n* (%)	573(40.99)	290(38.75)	283(43.79)	0.0577
Ethnicity, *n* (%)				
Non-Hispanic white	824(82.62)	475(85.02)	349(79.63)	0.0112
Non-Hispanic black	214(9.24)	95(7.2)	119(11.8)	
Mexican American/others	191(8.13)	87(7.78)	104(8.57)	
Body mass index (kg/m^2^)	29.04(28.61–29.47)	26.72(26.21–27.23)	31.94(31.24–32.63)	<0.0001
Laboratory data				
Glycohemoglobin (%)	5.96(5.88–6.03)	5.68(5.62–5.74)	6.3(6.18–6.42)	<0.0001
Fasting plasma glucose (mg/dl)	109.75(107.05–112.46)	96.88(95.2–98.56)	125.8(121.11–130.49)	<0.0001
HDL cholesterol (mg/dL)	53.06(52.06–54.06)	57.07(55.59–58.54)	48.07(46.57–49.57)	<0.0001
Total cholesterol (mg/dL)	193.47(190.44–196.51)	196.97(192.15–201.79)	189.11(184.4–193.83)	<0.0001
Triglycerides (mg/dL)	149.91(143.58–156.24)	125.34(120.23–130.44)	180.55(169.83–191.26)	<0.0001
Daily food intake				
Macronutrients				
Daily energy intake (kcal)/kg	21.82(21.19–22.46)	23.65(22.64–24.66)	19.54(18.7–20.37)	<0.0001
Daily carbohydrate intake ratio (%)	50.15(49.29–51.01)	51.25(50.26–52.25)	48.77(47.55–49.98)	<0.0001
Daily protein intake (gm)/kg	0.85(0.82–0.89)	0.9(0.85–0.94)	0.8(0.75–0.84)	<0.0001
Daily protein intake ratio (%)	16.06(15.62–16.49)	15.68(15.18–16.19)	16.52(15.89–17.16)	<0.0001
Daily total fat intake ratio (%)	33.8(33.18–34.41)	33.06(32.27–33.86)	34.71(33.75–35.67)	<0.0001
Total saturated fatty acids (gm)	20.98(20.15–21.81)	20.13(18.94–21.32)	22.04(20.64–23.44)	<0.0001
Total monounsaturated fatty acids (gm)	23.2(22.27–24.12)	22.25(20.88–23.61)	24.38(22.8–25.95)	<0.0001
Total polyunsaturated fatty acids (gm)	13.92(13.36–14.49)	13.62(12.78–14.45)	14.31(13.29–15.33)	<0.0001
Daily cholesterol intake (mg)	241.11(227.22–255)	219.72(203.13–236.32)	267.78(244.03–291.53)	<0.0001
Micronutrients				
Daily fiber intake (gm)	13.82(13.22–14.42)	14.06(13.18–14.94)	13.52(12.67–14.37)	<0.0001
Total folate (mcg)	340.79(328.33–353.26)	347.07(328.95–365.19)	332.97(314.62–351.31)	<0.0001
Vitamin B6 (mg)	1.69(1.62–1.76)	1.7(1.6–1.81)	1.67(1.55–1.79)	<0.0001
Vitamin B12 (mcg)	4.66(4.34–4.98)	4.6(4.15–5.05)	4.74(4.24–5.23)	<0.0001
Vitamin C (mg)	78.24(72.81–83.67)	78.18(71.14–85.22)	78.31(69.35–87.26)	<0.0001
Calcium (mg)	760.8(731.17–790.44)	745.21(704.7–785.73)	780.24(731.9–828.58)	<0.0001
Phosphorus (mg)	1083.49(1050.23–1116.74)	1055.72(1008.52–1102.91)	1118.1(1059.64–1176.57)	<0.0001
Magnesium (mg)	238.5(230.36–246.64)	239.31(227.53–251.09)	237.49(225.38–249.61)	<0.0001
Iron (mg)	13.57(13.03–14.1)	13.68(12.93–14.44)	13.42(12.62–14.21)	<0.0001
Zinc (mg)	9.99(9.49–10.49)	9.79(9.08–10.49)	10.25(9.58–10.92)	<0.0001
Copper (mg)	1.08(1.03–1.13)	1.06(1–1.13)	1.1(1.03–1.17)	<0.0001
Sodium (mg)	2782.85(2687.8–2877.9)	2670.51(2541.28–2799.74)	2922.9(2765.66–3080.13)	<0.0001
Potassium (mg)	2366.12(2298.49–2433.75)	2339.97(2242.93–2437.01)	2398.72(2271.46–2525.97)	<0.0001
Selenium (mcg)	88.69(85.61–91.78)	87.21(83.2–91.22)	90.55(85.22–95.88)	<0.0001
Caffeine (mg)	144.84(131.87–157.81)	140.78(125.71–155.85)	149.9(131.22–168.58)	<0.0001
Theobromine (mg)	30.68(26.29–35.07)	31.15(25.51–36.78)	30.09(24.48–35.71)	<0.0001
Alcohol (gm)	4.05(3.12–4.99)	5.13(3.73–6.52)	2.72(1.64–3.8)	<0.0001
Smoking, *n* (%)	606(49.67)	322(50.46)	284(48.68)	0.6232
Cardiovascular disease, *n* (%)	188(16.48)	92(14.96)	96(18.37)	0.2434
Hypertension, *n* (%)	882(70.82)	423(62.88)	459(80.72)	<0.0001
Diabetes mellitus, *n* (%)	425(30.81)	126(16.51)	299(48.63)	<0.0001

**Table 2 nutrients-14-02832-t002:** The area under curve (AUC) of the receiver operating characteristic (ROC) curve, accuracy, sensitivity, specificity, positive predictive value (PPV), and negative predictive value (NPV) from different models.

29 Features	XGboost	Random Forest	Logistic Regression	Deep Neural Network
Validation				
AUC of ROC	0.8	0.83	0.81	0.82
Accuracy	0.72	0.77	0.72	0.75
Sensitivity	0.72	0.72	0.67	0.74
Specificity	0.72	0.81	0.77	0.75
PPV	0.69	0.77	0.72	0.72
NPV	0.75	0.77	0.73	0.77
Test				
AUC of ROC	0.78	0.77	0.76	0.76
Accuracy	0.7	0.7	0.68	0.69
Sensitivity	0.66	0.6	0.63	0.66
Specificity	0.73	0.78	0.73	0.72
PPV	0.69	0.7	0.67	0.67
NPV	0.71	0.69	0.69	0.71

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
