# Peer review of "Development and Validation of an Insulin Resistance Model for a Population with Chronic Kidney Disease Using a Machine Learning Approach"

_nutrients, 2022, doi:10.3390/nu14142832_

Round 1
Reviewer 1 Report
The authors presented a well-conducted analysis concerning insulin resistance in CKD by utilizing the NHANES data with the aid of machine learning. Overall the study is methodologically sound and the results, discussion, and conclusion are appropriately reported. A few minor flaws can be detected and should be addressed:
1) Title: A shorter title would be preferable, such as "Development and Validation of an Insulin Resistance Model for a population with Chronic Kidney Disease using a machine learning approach".
2) Abstract: I suggest changing the phrase "we recruited" since you did not perform the actual recruiting. Maybe use passive voice, such as "in this study, 71,916 participants were enrolled". Make adjustments throughout the text.
3) Abstract: Change the (n=1,229) since you already mentioned it in the previous sentence.
4) Introduction: "While the estimated glomerular filtration...IR gradually progresses". These 2 sentences appear unnecessary at this point.
5) Introduction: IS in the last paragraph should probably be IR.
6) Section 2.2.: Provide a reference for the used CKD-EPI equation.
7) Section 2.3.: How was the cutoff for IR set at >3? While information is provided about the different cutoffs in previous studies, I cannot find an explanation for the authors using an arbitrary cutoff.
8) Section 2.3.: Change OMA-IR to HOMA-IR and incidences to incidence.
9) Section 2.5.: Please mention if the continuous variables were tested for normality, how they are presented, and through which tests they were analyzed in the initial descriptive statistics of the results.
10) Results: Table 1 is never cited in the text. Please provide more explanatory figure legends where possible.
11) Discussion: What are the clinical implications of the study? What is the next step?
Author Response
The authors presented a well-conducted analysis concerning insulin resistance in CKD by utilizing the NHANES data with the aid of machine learning. Overall the study is methodologically sound and the results, discussion, and conclusion are appropriately reported. A few minor flaws can be detected and should be addressed:
- Title: A shorter title would be preferable, such as "Development and Validation of an Insulin Resistance Model for a population with Chronic Kidney Disease using a machine learning approach".
àThanks for this comment. We shorten our title as your suggestion.
- Abstract: I suggest changing the phrase "we recruited" since you did not perform the actual recruiting. Maybe use passive voice, such as "in this study, 71,916 participants were enrolled". Make adjustments throughout the text.
àThanks for this comment. We revised it accordingly.
3) Abstract: Change the (n=1,229) since you already mentioned it in the previous sentence.
àWe deleted it according your suggestion.
4) Introduction: "While the estimated glomerular filtration...IR gradually progresses". These 2 sentences appear unnecessary at this point.
àThanks for this comment. We deleted these 2 sentences accordingly.
5) Introduction: IS in the last paragraph should probably be IR.
àThanks for this comment. We revised it accordingly.
6) Section 2.2.: Provide a reference for the used CKD-EPI equation.
àThanks for this comment. We revised it accordingly.
7) Section 2.3.: How was the cutoff for IR set at >3? While information is provided about the different cutoffs in previous studies, I cannot find an explanation for the authors using an arbitrary cutoff.
àThanks for this comment. Because of the cutoff value of IR set in patients with CKD is still without consensus (range 1.23-5.65), we set the cutoff value of IR in this study as >3. HOMA-IR index rose progressively with age, plateaued between age 13 and 15 years and started decreasing afterward. HOMA-IR peaked at age 13 years in girls and at 15 years in boys. If HOMA-IR cutoff point is 3.02, the AUROC = 0.73, 95 % CI = 0.70–0.75; sensitivity = 46.3 %; and specificity = 82.6 % (Acta Diabetol. 2016 Apr;53(2):251-60). We cited this reference and added more explanation in the text.
8) Section 2.3.: Change OMA-IR to HOMA-IR and incidences to incidence.
àThanks for this comment. We revised it accordingly.
9) Section 2.5.: Please mention if the continuous variables were tested for normality, how they are presented, and through which tests they were analyzed in the initial descriptive statistics of the results.
àThanks for this comment. Because NHANES dataset is a multiple complex survey design, we presented the weighted difference mean and percentage in basic characteristics to represent sample-weighted data of whole population. However, we used original unweighted data to perform model building process of machine learning and deep learning. Therefore, we added the unweighted data of descriptive statistics in Supplementary table 2. For unweighted data (Supplementary table 2), continuous variables are reported as means ± SD and categorical data as numbers (percentages). Differences in clinical variables between insulin resistance status were tested by using the Chi-square test for categorical variables or paired t test for continuous variables. All reported p-values were two-sided and considered significant with p <0.05. We also revised our statistical method.
10) Results: Table 1 is never cited in the text. Please provide more explanatory figure legends where possible.
à We cited the table 1 in section 3.1. Also, we provide more explanatory figure legends.
11) Discussion: What are the clinical implications of the study? What is the next step?
àThanks for this comment. We added some explanation in this part. We used data from clinical practice via machine learning to find out a good algorithm to predict IR. The IR in CKD is strongly associated with atherosclerosis. After this study, we can have early prediction of IR in patients with CKD to avoid further CVD and functional deteriorations. In the future, first, we will validate this algorithm in other large database via federated learning. Second, we will further correlate the association of IR in CKD to clinical outcomes, including CVD, renal function deterioration, and all-cause mortality via machine learning. We added the above explanation in the text.
Reviewer 2 Report
The study entitled “Development and Validation of an Insulin Resistance Model for a Population with Chronic Kidney Disease: Using a Machine-learning Approach Using the National Health and Nutrition Examination Survey, USA, 1999-2012 by Lee et al., is well designed and performed, and results are clearly presented in the manuscript. This study will help to determine risk factors and predict Insulin Resistance (IR) in patients with chronic kidney disease (CKD) using machine learning (ML) algorithms, mainly RF (random forest). I appreciate the author’s efforts in making this manuscript relevant to clinical importance.
Author Response
àThanks for this comment.